# RNA-Seq Analysis Demystify the Pathways of UV-A Supplementation in Different Photoperiods Integrated with Blue and Red Light on Morphology and Phytochemical Profile of Kale

**DOI:** 10.3390/antiox12030737

**Published:** 2023-03-16

**Authors:** Haozhao Jiang, Yamin Li, Jiehui Tan, Xinyang He, Shijun Zhu, Rui He, Xiaojuan Liu, Houcheng Liu

**Affiliations:** College of Horticulture, South China Agricultural University, Guangzhou 510642, China

**Keywords:** UV-A, kale, transcriptomics, growth, glucosinolates

## Abstract

As an indispensable element in the morphology and phytochemical profile of plants, UV-A has proved to help promote the growth and quality of kale. In this study, UV-A supplementation in different photoperiods (light period supplemental UVA = LS, dark period supplemental UVA = DS, and light-dark period supplemental UVA = LDS) contributed to yielding greater biomass production (fresh weight, dry weight, and plant moisture content), thus improving morphology (plant height, stem diameter, etc.) and promoting higher phytochemicals content (flavonoids, vitamin c, etc.), especially glucosinolates. To fathom its mechanisms, this study, using RNA-seq, verified that UV-A supplementation treatments signally generated related DEGs of plant hormone signal pathway, circadian rhythm plant pathway, glucosinolate pathway, etc. Moreover, 2047 DEGs were obtained in WGCNA, illustrating the correlations between genes, treatments, and pathways. Additionally, DS remarkedly up-regulated related DEGs of the key pathways and ultimately contributed to promoting the stem diameter, plant height, etc., thus increasing the pigment, biomass, vitamin c, etc., enhancing the antioxidant capacity, and most importantly, boosting the accumulations of glucosinolates in kale. In short, this study displayed new insights into UV-A supplementation affected the pathways related to the morphology and phytochemical profile of kale in plant factories.

## 1. Introduction

Light is an indispensable factor that affects all aspects of plant growth by providing energy to plants. Plants could sense light signals through different receptors, including phytochromes (*PHYA-PHYE*) that sense red and far-red light, cryptochromes (*CRY 1*, *CRY 2*, and *CRY DASH*) that sense blue light and UVA, and phototropin (*PHOT1* and *PHOT2*) and UVB-sensing *UVR8*, induce extensive transcriptional reprogramming in plants to optimize plant growth, development, and stress response [1,2].

The reduction of ultraviolet light (UVB and UVA) leads to higher growth, biomass, and pigment content in monocots such as sorghum and wheat [3,4]. Supplemental UV-A (0.159 mW·cm^−2^) with PAR (photosynthetically active radiation, 35 µmol·m^−2^ s^−1^) signally expanded the rosettes diameter of *Arabidopsis thaliana* [5]. Lettuce under 237 μmol·m^−2^ s^−1^ RBFR shoot dry weight was increased by 27%, 29%, and 15% in the UVA-10 (10 µmol·m^−2^ s^−1^), UVA-20 (20 µmol·m^−2^ s^−1^), and UVA-30 (30 µmol·m^−2^ s^−1^) treatments, respectively, which correlated with 31% (UVA-10), 32% (UVA-20), and 14% (UVA-30) larger leaf areas [6]. Supplemental 8 and 16 h UV-A (2.28 W·m^−2^, 369 nm) under 9R1B = 220 μmol·m^−2^ s^−1^ stimulated plant biomass production in tomato seedlings by 29% and 33%, respectively, mainly due to larger leaves (i.e., 22% and 31% in 8 and 16 h UV-A, respectively), which facilitated light capture, while 8 h UV-A reached the biggest stem width [7].

Chloroplast movement could be induced by an increased light intensity-to-blue light ratio, a process mediated by the phototropin (UVA/blue light receptor)-related *NPL1* gene that controls chloroplast relocation [8,9]. UVA exposure (9.47 W·m^−2^) caused increments in lutein (by 22.4%), chlorophyll b (by 30.7%), neoxanthin (by 33.5%), and chlorophyll a (by 67%) in broccoli sprouts [10]. Day supplemental UV-A (31.2 µmol·m^−2^ s^−1^) slightly increased the root and total biomass (4.9% and 5.3%) of pea seedlings and significantly decreased the shoot chlorophyll b of pea seedlings content (19.4% and 19.4%) [11]. Supplemental low-intensity (10 µmol·m^−2^ s^−1^) UV-A clearly improved the content of vitamin C, chlorophyll, and carotenoid, while higher intensity (40 µmol·m^−2^ s^−1^) UV-A caused lower chlorophyll content in lettuce compared with control (230 μmol·m^−2^ s^−1^ RB + 7 μmol·m^−2^ s^−1^ FR) [12].

As a functional vegetable with high antioxidant capacity, kale (*Brassica oleracea* var. *sabellica*) is rich in healthy functional substances, such as vitamin C, phenolic compounds, and flavonoids, especially glucosinolates [13], and generates more zest worldwide.

Different light could trigger different photoreceptors in plants, including phytochromes (*PHYA-PHYE*) that sense red and far-red light, cryptochromes (*CRY 1*, *CRY 2* and *CRY DASH*) that sense blue light and UV-A, and Phototin (*PHOT1* and *PHOT2*), and UVB-sensing *UVR8* that induces extensive transcriptional reprogramming in plants to optimize plant growth, development, and stress response [2]. One of the most important and widespread protective responses of plants against UV radiation is the induction and synthesis of flavonoids and related phenolic compounds, which are UV-shielding ingredients and antioxidants [14]. Glucosinolates and their degradation products are known to play important roles in plant interaction with herbivores and microorganisms [15]. Aliphatic glucosinolate biogenesis genes (*CYP79F1*, *CYP83A1*, *UGT74B1*, and *AOP2*) were up-regulated by yellow (570 nm), blue (455 nm), and purple (420 nm) LED lights, which induced the accumulation of glucoraphanin and glucosinolates (3.43-fold, 2.09-fold, and 3.66-fold) [16].

In recent years, RNA-seq has emerged as a powerful technique for analyzing gene expression in response to specific stimuli in a wide range of biological systems. RNA-seq has been used to investigate the overall expression profile of plants under different stresses, revealing the signal transduction pathways involved in the resistance network [17]. There are many studies that revealed the effects of different treatments on vegetable growth and development, functional phytochemicals, and stress resistance, such as radish [18], cabbage [19], and purple broccoli [20].

The growth of vegetables in plant factories is mainly affected by light. Although extensive research has been carried out on the effects of UV-B, red, and blue light on plants, restricted studies existed on the comprehensive effects of UV-A with red and blue light on kale’s morphology and quality, while UV-A also has an important impact on plants [21]. In this study, RNA-seq technology was applied to decode the effects of supplemental UV-A in different photoperiods on the morphology and quality in kale under red and blue light in an artificial lighting plant factory.

## 2. Materials and Methods

### 2.1. Plant Materials, Growth Condition, and Light Treatments

This experiment took place at South China Agricultural University’s artificial lighting plant factory. Kale (*Brassica oleracea* var. *sabellica*) cv ‘Jingyu No.2’ (Beijing JingYan YiNong Sci-Tech Development Center, Beijing, China) seeds were sown in moist sponge blocks and maintained in a dark germination chamber for 2 d. Then, the germinated seeds were kept in a deep flow technique system with 1/2 Hoagland solution, as well as with 400–600 μmol·mol^−1^ CO_2_, 21 ± 2 °C temperature, EC ≈ 1.8 mS·cm^−1^, pH ≈ 6.4, 55–60% relative humidity, and 250 μmol·m^−2^ s^−1^ PPFD white LED lighting from 8:00 to 18:00. Seedlings with 2 expended true leaves were transplanted into the planting plate (90 cm × 60 cm, 24 plants/plate) after 2 weeks.

The LED panels (Chenghui Equipment Co., Ltd., Guangzhou, China; 150 cm × 30 cm) with blue (460 ± 10 nm), red (660 ± 10 nm), and supplemental UV-A (380 ± 10 nm) LEDs were applied. The kale seedlings were cultured under 4 treatments with primary light (red: blue = 1:1 at PPFD of 250 μmol·m^−2^ s^−1^): CK (primary light, non-UVA treated) after transplantation. The photoperiod of UV-A supplementation (12 μmol·m^−2^ s^−1^) for each treatment was: light period supplemental UVA (6:00–18:00, LS), dark period supplemental UVA (18:00–06:00, DS), and light-dark period supplemental UVA (12:00–00:00, LDS). Samples for RNA-seq were collected at 10 d after UV-A supplementation. The other samples for biometric measurements and quality assays were collected at 21 d after UV-A supplementation. All the fresh samples for quality assays have been ground and stored at −20 °C.

### 2.2. Biometric Measurements

About 7 kale plants in each treatment were stochastically selected for measurement of morphology indexes and fresh weight, combined at 105 °C for 2 h with 70 °C for 72 h to determine the dry weight and calculate the moisture content of samples. The moisture content of kale samples (%) = (FW − DW)/FW × 100%.

### 2.3. Pigment Content Assay

Fresh leaves of kale (0.2 g) were chopped and soaked in 6.0 mL of acetone ethanol mixture (acetone: ethanol = 1:1, *v*:*v*) and stored in darkness for 24 h. The extract solution absorbances were determined by UV spectrophotometer (Shimadzu UV-16A, Shimadzu, Corporation, Kyoto, Japan) at 663 nm (A663), 645 nm (A645), and 440 nm (A440). The pigment contents were determined according to Lichtenthaler [22], as follows:

Chl a content (mg/g FW) = (12.70 × A663 − 2.69 × A645) × 6 mL/(1000 × 0.2 g);

Chl b content (mg/g FW) = (22.90 × A645 − 4.86 × A663) × 6 mL/(1000 × 0.2 g);

Chl a + Chl b content (mg/g FW) = (8.02 × A663 + 20.20 × A645) × 6 mL/(1000 × 0.2 g);

Carotenoid content (mg/g FW) = (4.70 × A440 − 2.17 × A663 − 5.45 × A645) × 6 mL/(1000 × 0.2 g).

### 2.4. Phytochemical Measurements

#### 2.4.1. Soluble Protein Content Assay

The soluble protein content was determined by Coomassie blue staining [23]. Combined stored samples (0.5 g) with 8 mL distilled water and then extracted the supernatant. The supernatant (0.5 mL) plus distilled water was well-mixed with Coomassie brilliant blue G-250 solution and measured at 595 nm by a UV spectrophotometer 5 min later.

#### 2.4.2. Soluble Sugar Content Assay

Soluble sugar content was determined by anthrone colorimetry [24]. Frozen samples were well-mixed with distilled water, after repetitive 100 °C water baths. Later, the filtered solutions were well-mixed in the sequence of distilled water, anthrone ethyl acetate reagent (Sinophaem, Beijing, China), and concentrated sulfuric acid. The supernatant was determined at 625 nm by UV spectrophotometer.

#### 2.4.3. Vitamin C Content Measurement

Vitamin C content was determined by molybdenum blue spectrophotometry [25]. Samples were homogenized with oxalic acid ethylene diamine tetraacetic acid solution (*w*/*v*) and then filtered. The supernatants were mixed in the sequence of partial phosphoric acid-acetic acid solution (*w*/*v*), sulfuric acid solution (*v*/*v*), and ammonium molybdate solution (*w*/*v*) and then measured at 705 nm by a UV spectrophotometer.

#### 2.4.4. Nitrate Content Measurement

Samples soaked in distilled water were heated and filtered. Then, the solution was mixed with 5% salicylic, sulfuric acid, and 8% NaOH sequentially. The nitrate content was determined using a UV spectrophotometer at 410 nm [26].

#### 2.4.5. DPPH Radical Inhibition Percentage Measurement

The DPPH radical inhibition percentage (DPPH) measurement was determined according to Musa et al. [27]. Samples soaked in ethanol were stored in darkness for 30 min. With three types of mixtures (Aj: supernatant mixed with ethanol; Ai: supernatant mixed with DPPH; Ac: DPPH mixed with ethanol) prepared, DPPH was determined at 517 nm by the UV spectrophotometer.

#### 2.4.6. Ferric Ion-Reducing Antioxidant Power Measurement

The Ferric ion-reducing Antioxidant Power (FRAP) assay was determined according to Benzie and Strain [28]. Supernatants were well-mixed with TPTZ solution, then incubated at 37 °C for 10 min and determined at 593 nm.

#### 2.4.7. Total Phenolic Content Measurement

The total phenolic content was measured according to Rahman [29]. Supernatants (samples soaked with methanol) were treated in the sequence of 0.5 mL Folin–Ciocalteu’s phenol, 1.5 mL 26.7% Na_2_CO_3_, and 7 mL distilled water and measured at 760 nm.

#### 2.4.8. Total Flavonoids Content Measurement

The total flavonoids content was measured by using the Al(NO_3_)_3_ colorimetric assay [30]. The sample extract (5 mL) was mixed with methanol and NaNO_2_ solution and added to 10% AlCl_3_ (0.35 mL) and 5 mL 5% NaOH in sequence. The sample absorbance was measured at 510 nm.

#### 2.4.9. Mineral Element Contents Measurement

The mineral element (nitrogen (N), phosphorus (P), Kalium (K), calcium (Ca), magnesium (Mg), sulfur (S), and zinc (Zn)) contents measurement was based on Gao [31]. Mineral element accumulation contents (mg/per plant) = mineral element content (g·kg^−1^ DW) × dry weight per plant (kg DW) × 1000.

#### 2.4.10. Glucosinolates Content Measurement

Glucosinolates were extracted and determined according to Li [32]. The frozen-dried samples were extracted with methanol and then purified and desulfurized with the ion-exchange method. The glucosinolates were separated and identified using high-performance liquid chromatography (HPLC, Waters Alliance e2695). With a 5 μm C18 column (Waters, 250 mm length, 4.6 mm diameter) for glucosinolate separation, elution through mobile phase A (water, 18.2 MΩ·cm resistance), and mobile phase B (acetonitrile), the optimum column temperature was 30 °C. The detector monitored glucosinolates at 229 nm. The gradient conditions were set as follows: 0 to 32 min (100% solvent A volume), 32 to 38 min (80% solvent A volume), and 38 to 40 min (100% solvent B volume). The individual glucosinolates were identified according to their HPLC retention times and quantified with sinigrin (Sigma-Aldrich, St. Louis, MO, USA), which was used as an internal reference substance with their HPLC area and relative response factors (ISO 9167-1,1992).

### 2.5. RNA Extraction and Illumina Sequencing

The RNA libraries were sequenced on the Illumina sequencing platform by Genedenovo Biotechnology Co., Ltd. (Guangzhou, China); the following detailed experimental methods were provided by the company. The total RNA was extracted from the leaves by Total RNA Isolation Extraction Kit (Vazyme Biotech Co., Ltd., Nanjing, China). After total RNA extraction, the integrity of the RNA was meticulously assessed using the Agilent 2100 bioanalyzer. The messenger RNA (mRNA) was selected and enriched by polyA. The enriched mRNA was subsequently fragmented and reversely transcribed into complementary DNA (cDNA) using NEBNext Ultra RNA Library Prep Kit for Illumina (NEB#7530, New England Biolabs, Ipswich, MA, USA). The purified double-stranded cDNA fragments were end-repaired. A base was added and ligated to Illumina sequencing adapters. The libraries were sequenced by applying Illumina HiseqTM 4000 (Illumina, San Diego, CA, USA). A total of 12 groups (3 replicates per treatment) of RNA extraction were applied to the RNA-seq analysis for this study. The accession number of Sequence Read Archive (SRA) is PRJNA940045 (https://www.ncbi.nlm.nih.gov/sra/PRJNA940045, accessed on 4 March 2023).

### 2.6. Mapping, DEGs, GO Enrichment, and KEGG Pathway Analysis

Through fastp [33], the quality control of the raw reads from the computer was performed, and the low-quality data was filtered to obtain clean reads. The clean reads were mapped to the *Brassica oleracea* var. *oleracea* (wild cabbage) reference genome sequence (https://www.ncbi.nlm.nih.gov/assembly/GCF_000695525.1, accessed on 4 April 2021). The gene expression level was calculated using the fragments per kilobase of transcript per million mapped reads (FPKM). This study used DESeq2 [34] to detect differential expression genes (DEGs). Genes with |log2FC| > 1 and FDR < 0.05 were defined as signally differential expression genes (DEGs). The DEGs of kale under different light/dark photoperiod treatments were subjected to GO and KEGG functional enrichment analysis.

### 2.7. Weighted Gene Co-Expression Network (WGCNA) Analysis and Gene Network Visualization

A total of 2047 DEGs were imported to construct co-expression networks, according to the gene expression matrix by using the WGCNA (version 1.47) package in R [35], and construct co-expression modules using the automatic network construction function block-wise modules to determine the gene expression values with default settings, except that TOM type was unsigned, the power was 8, merge Cut Height was 0.1, and the min module size was 50. Genes were clustered into 8 correlated modules.

### 2.8. qRT-PCR Analysis

A total of 15 DEGs were stochastically selected to verify the expression profiles established by RNA-seq. qRT-PCR was applied by SYBR Premix Ex Taq II (Tli RNaseH Plus) (Takara Bio, Dalian, China), and the reactions took place on a LightCycler 480 Real-Time PCR system (Roche, Basel, Switzerland). Primer 5.0 software was used to design the primers. The expression data were analyzed by using the 2^−ΔΔCT^ method [36], with the *ACT* gene for normalization. Primers used for this study could be found in Appendix A.

### 2.9. Statistical Analysis

The measurements were calculated with three replications per treatment, using SPSS 23.0 software (SPSS Inc., Chicago, IL, USA) for statistical analysis and applying analysis of variance (ANOVA), followed by Duncan’s test for Significance determination among the treatments. All figures were elaborated by TBtools software [37] and OriginPro 9.0 software (OriginLab Inc., Northampton, UK).

## 3. Results

### 3.1. Morphology and Biomass of Kale in UV-A Supplementation

Supplemental UV-A in different photoperiods massively affected the morphology and biomass of kale (Appendix A).

Compared with CK, dark-UVA supplementation (DS), and light/dark-UVA supplementation (LDS) distinctly promoted the stem diameter and total leaf area of kale, with 102.38%, 57.14%, and 100.38%, 80.89%, respectively, while the light-UVA supplementation (LS) barely changed. DS and LDS presented higher specific leaf weight than CK, by ~27.67% and ~17.87%, which indicated thicker leaves than CK, while LS showed no difference. UV-A supplementation (LS, DS, and LDS) signally increased plant height of kale plants and yielded higher shoot fresh weight, with increments of 45.16%, 98.39%, 45.70%, 56.29%, 191.82%, and 84.49%, respectively, in accordance with the results obtained on the dry weight of shoot and root (40.71%, 159.11%, 54.64%, 6.87%, 40.38%, and 19.23%) of fresh weight. With biomass results revealed, UV-A supplementation (LS, DS, and LDS) signally increased the plant moisture content in kale, with 2.12%, 2.94%, and 2.77%, respectively. Additionally, dark-UVA supplementation (DS) reached the maximum morphology and biomass of kale among all the UV-A treatments.

In short, dark-UVA supplementation (DS) and light/dark-UVA supplementation (LDS) prominently improved morphology and yielded higher biomass of kale.

### 3.2. Pigment of Kale Leaves in UV-A Supplementation

The pigment biosynthesis of kale leaf was found to be markedly increased in UV-A supplementation at different photoperiods, which ultimately resembled the SPAD (Appendix A). LS, DS, and LDS exhibited higher contents of chlorophyll a and b, with increments of 7.95%, 9.09%, 9.09%, and 7.69%, 15.38%, and 17.95%, respectively. The lower chlorophyll a/b values observed in shade plants can be attributed to the higher absorption of blue-violet light by chlorophyll b, compared to chlorophyll a, which is more prominent in sunny plants. Therefore, shade plants can effectively utilize the predominant blue-violet diffuse light in shady conditions. DS and LDS observably lowered the chlorophyll a/b (7.08% and 8.41%). Meanwhile, the carotenoid exerted the highest content in LS (~5.89%), while DS and LDS showed no noteworthy difference. Compared to CK, DS indicated the highest SPAD (~6.29%).

Hence, dark-UVA supplementation (DS) and light/dark-UVA supplementation (LDS) signally boosted the pigment content of kale and ultimately accumulated the biomass.

### 3.3. Soluble Protein, Soluble Sugar, Vitamin C, and Nitrate Contents Assay

Supplementing UV-A in different photoperiods signally affected the nutritional content of kale. UV-A supplementation in different photoperiods (LS, DS, and LDS) noteworthily increased the contents of soluble protein and vitamin C of kale, while the soluble sugar content remained unchanged, with massive increases of 21.36%, 24.36, 25.70%, and 20.65%, 26.09%, 27.74%, respectively (Appendix A). However, an arresting reduction of nitrate content was observed in UV-A supplementation (LS, DS, and LDS) with 20.44%, 13.14%, and 16.06%, respectively.

Overall, UV-A supplementation in different photoperiods exerted higher nutritional quality of kale, while DS indicated the maximum.

### 3.4. Antioxidant Capacity and Compounds Assay

UV-A supplementation in different photoperiods markedly affected the antioxidant capacity and compounds in kale (Appendix A). DS and LDS signally promoted FRAP of kale, while no remarkable differences were observed in the DPPH and total flavonoids of kale under UV-A supplementation. However, LS treatment revealed a conspicuous reduction in total phenolic content.

### 3.5. Mineral Element Content Assay

UV-A supplementation in different photoperiods massively promoted the mineral elements accumulation in the shoot of kale, while different mineral elements responded differently towards different UV-A treatments (Appendix A). UV-A supplementation in different photoperiods (LS, DS and LDS) yielded higher contents of Ca (12.61%, 10.87% and 8.2%), Mg (5.04%, 9.83% and 10.31%), and S (27.08%, 26.48% and 22.58%), while N content barely changed. DS and LDS prominently increased the P contents (14.71% and 11.46%), while LS significantly increased the Zn content (10.51%).

UV-A supplementation (LS, DS, and LDS) highly increased the mineral elements accumulation per plant by combining the shoot dry weight of kale (Appendix A). Supplemental UV-A in different photoperiods massively boosted the accumulation of all the mineral elements contents: N (47.78%, 177.56%, and 63.40%), P (39.46%, 196.91%, and 72.53%), K (41.18%, 142.29%, and 40.98%), Ca (59.06%, 186.94%, and 67.76%), Mg (48.65%, 184.98, and 71.17%), Zn (56.15%, 161.54, and 52.31%), and S (79.55%, 227.29%, and 90.06%), while DS reached the maximum.

### 3.6. Glucosinolate Content Assay

Seven individual glucosinolates were extracted and detected by HPLC in this study (Figure 1). Among them, there are three aliphatic GLSs: Sinigrin (SIN), Gluconapin (GNA), and Glucobrassicanpin (GBN), and four indolic GLSs: 4-Hydroxyglucobrassicin (4-OH), Glucobrassicin (GBS), 4-Methoxyglucobrassicin (4ME) and Neoglucobrassicin (NEO). The total indolic GLSs content in kale exerted predominantly higher than total aliphatic GLSs, accounting for 98.7% of total GLSs on aggregate. Additionally, GBS reached the highest proportion (66.22%), with NEO (30.75%) following.

UV-A supplementation in different photoperiods (LS, DS, and LDS) markedly boosted the content of indolic GLSs in kale, with 54.12%, 100.66%, and 131.09%, respectively. Among them, DS treatment indicated the highest 4ME content by ~183.33%, while LDS treatment revealed the highest GBS content by ~165.13%. LS yielded higher SIN content (140%) and DS exerted higher GNA content (80%), while total aliphatic GLSs content remained no distinct difference.

Overall, supplemental UV-A conspicuously increased GLSs content of kale, while DS and LDS treatments displayed higher.

### 3.7. Heatmap and Multivariate Principal Component Analysis

A heatmap presented an integrated overview of the impact of different supplemental UV-A treatments on the morphology and quality of kale (Figure 2).

The cluster exhibited different morphology and quality of kale in different photoperiod UV-A supplementations and CK at harvest. The kale in CK exerted higher content of chlorophyll a/b, nitrate, and K, while UV-A supplementation massively revealed higher pigment content, fresh weight, dry weight, plant moisture content, indolic GLSs, 4OH, 4ME, GNA, S, P, Mg, total phenolic, soluble sugar, soluble protein, and Vc, etc. These results verified supplemental UV-A treatments enhanced the growth and promoted the phytochemical accumulation of kale.

The principal component analysis (PCA) was performed to compare the relevance of all quality characteristics of kale between supplemental UV-A treatments and CK (Figure 3). The first 10 principal components, PC1–PC10 (eigenvalues > 1), account for 99.43% of the cumulative variance.

Figure 3 showed that presenting the first two factors (PC1 vs. PC2) revealed 63.01% of kale’s total variance in supplemental UV-A treatments and CK. The results indicated a clear separation between CK and LS, DS and LDS, validating significant differences between the results obtained between CK and the supplemental UV-A treatments. The results indicated the relationship between morphology and quality by confirming the angle between two vectors (0° < positively correlated < 90°; uncorrelated, 90°; 90° < negatively correlated < 180°). There were strong positive correlations between PH, SD, TFW, PMC, TLA, SLA, SIN, DPPH, GBN, Ca, NEO, S, 4OH, 4ME, GBS, Mg, FRAP, soluble sugar, soluble protein, indolic GLSs, and so forth, as their angles were less than 90°.

### 3.8. RNA-Seq Analysis

#### 3.8.1. Illumina Sequencing, Mapping Reads, and Transcript Identification

In this study, an Illumina HiSeq platform was used for sequencing 12 kale samples, with a total of 77.6 Gb of clean data generated. Appendix A displays more than 97.62% of clean reads were >Q20, 93.23% of clean reads were >Q30, with GC ranging from 47.54% to 47.97% (with an average of 47.71%), and more than 91.05% of these reads were mapped to the *Brassica oleracea* var. *oleracea* genome (GCF_000695525.1), which indicates that the high-quality sequencing and good assembly effect meet the needs of subsequent bioinformatics analysis. To calculate the gene expression values between each pair of samples, FPKM and Pearson correlation coefficients were utilized. The R^2^ between the two samples is >0.97, indicating great repeatability of the experiment and highly reliable sequencing results (Appendix A).

#### 3.8.2. Identification of Differential Expression Genes (DEGs)

Genes were considered differential expressions by using DESeq2 with the false discovery rate (FDR) < 0.05 and the |log2FC (fold change)| > 1. A total of 2047 DEGs have been identified, while there were more up-regulated DEGs in the CK vs. DS comparison and more down-regulated DEGs in the CK vs. LDS (Appendix A).

#### 3.8.3. Gene Ontology (GO) Enrichment and KEGG Pathway Analysis of DEGs

Gene Ontology (GO) provides a dynamically updated controlled vocabulary to comprehensively describe the properties of genes and gene products in organisms. The results of GO classification indicated that the DEGs were enriched in 23 biological processes (BPs), 17 cellular components (CCs), and 11 molecular functions (MFs). Compared to CK, a massive amount of up-regulated DEGs exhibited enrichment in DS, while LDS revealed the most down-regulated DEGs. Significant GO terms were enriched in the “cellular process”, “metabolic process”, “single-organism process”, “response to stimulus”, “biological regulation”, and “regulation of biological process” in the BP category. In the CC category, the top three enriched GO terms were “cell”, “cell part”, and “organelle”. In the MF category, the remarkedly enriched GO terms were “binding” and “catalytic activity” (Appendix A).

In organisms, different genes coordinately perform their biological functions. Pathway-based analysis helps to further understand the biological functions of genes. Determining significant enrichment can identify the most crucial biochemical metabolic and signal transduction pathways that are involved in differential genes.

Significant pathways (*p* < 0.05) were screened and utilized for comparative analyses. The top 20 KEGG pathways were shown in Appendix A. It revealed 486 DEGs that were enriched in 14 significant pathways, including “Biosynthesis of secondary metabolites” (n = 153, 31.48%) (mainly about ‘Biosynthesis of phytochemical compounds’, ‘Polyketide sugar unit biosynthesis’, ‘Plant terpenoid biosynthesis’, ‘Cofactor and vitamin metabolism’, etc.), “Plant hormone signal transduction” (n = 58, 11.93%), “Ribosome” (n = 53, 10.91%), “Plant-pathogen interaction” (n = 30, 6.17%), “Phenylpropanoid biosynthesis” (n = 28, 5.76%), “MAPK signaling pathway-plant” (n = 27, 5.56%), “Glutathione metabolism” (n = 17, 35%), “Cyanoamino acid metabolism” (n = 13, 2.67%), “alpha-Linolenic acid metabolism” (n = 13, 2.67%), “2-Oxocarboxylic acid metabolism” (n = 13, 2.67%), “Carotenoid biosynthesis” (n = 9, 1.85%), “ Circadian rhythm-plant” (n = 9, 1.85%), “Glucosinolate biosynthesis” (n = 8, 1.65%), and “Linoleic acid metabolism” (n = 4, 1.44%).

In short, these pathways might be the primary metabolic pathways involved in kale’s response to different UV-A supplementation.

#### 3.8.4. Identification of Key Regulatory Genes Involved in Important Pathways

The “biosynthesis of secondary metabolites”, “plant hormone signal transduction”, “Circadian rhythm-plant”, “Glucosinolate biosynthesis” pathways, and other pathways of photosynthesis were identified as being significantly enriched in kale.

The network of “plant hormone signal transduction” pathway revealed 58 DEGs significantly enriched (Figure 4). A total of 27 DEGs were identified in the Auxin pathway, with *SAUR22-like*, *SAUR20*, *SAUR24*, and *SAUR50* markedly down-regulated in LDS, *IAA27*, *IAA9*, and *SAUR50* massively up-regulated in DS, and *IAA17* up-regulated in CK and LS. In the CTK pathway, *AHP4*, *ARR8*, and *ARR5* were up-regulated in CK and LS and down-regulated in DS and LDS. In the GA pathway, the *GID1B* gene was significantly down-regulated in LDS and up-regulated in CK.

The supplemental UV-A in different photoperiods typically interact with the circadian rhythm pathway, with eight DEGs and seven constans-like DEGs markedly enriched (Appendix A). *APRR1*, *APRR5*, *LHY*, *TCP21*, and *PHYB* indicated signal up-regulation in DS, while *APPR9* inhibited expression. The series of constans-like genes perform similar functions during light treatment. *COL9*, *COL15*, *CIA2*, and *COL10* were all identified as observably up-regulated in DS. Above them, *COL-6* (ncbi_106343570, *CIA2*) indicated massively up-regulation in DS.

A total of eight DEGs encoding enzymes related to the glucosinolate biosynthesis pathway were identified (Appendix A). *CYP79A*, *BCAT4*, *SOT17*, *IMDH1*, and *MAM2* exerted dominant up-regulation in LDS. The other *CYP79A2* and *CYP79B1* displayed prominent up-regulation in DS, but arrested down-regulation in CK and LS. *CYP79B3* revealed distinctly up-regulation in CK and DS, while down-regulation in LS.

Thus, different light treatments might trigger specific photoreceptors, exhibit key regulatory genes (i.e., IAA9, COL6, and other related genes/proteins), transmit specific signals, and ultimately affect the growth and biomass of kale (Figure 5).

#### 3.8.5. WGCNA Analysis

To gain more in-depth analyses into the gene regulatory network of the growth and phytochemical quality of kale, this study engaged WGCNA to perform a co-expression network using 2047 DEGs with morphological and nutritional parameters (Figure 6A). DEGs were acutely partitioned into eight distinct co-expression modules (turquoise, blue, green, brown, black, red, pink, and magenta) (Figure 6B). The gene numbers of eight modules revealed MM.turquoise and MM.blue owned more DEGs, while MM.pink and MM.magenta showed less. Among them, the module-sample expression pattern proved that CK has positive correlations with MM.magenta and MM.red. LS treatment has positive correlations with MM.pink and MM.blue, while DS treatment indicated positive correlations with MM.black, MM.green, and MM.turquoise, apart from them. LDS treatment showed a positive correlation with MM.black and MM.brown (Figure 6C). Meanwhile, regarding the traits of morphology (PH, SD, TLA, SPAD, TFW, TDW, PMC, SLW, chla, chlb, caro), most of the mineral element content (N, P, Ca, Mg, and S), GLS content (GNA, 4OH, 4ME, NEO), and quality (TP, FRAP, SP, VC) were found to be positively associated with DEGs expression in black, brown, green, turquoise, and pink modules. Chla/b, K, and nitrate content showed a positive correlation with the magenta modules (Figure 6D). Additionally, the “plant hormone signal transduction” pathway was found to be significantly enriched in MM.pink and MM.blue. “Biosynthesis of secondary metabolites” and “phenylpropanoid” pathways were found to be highly enriched in MM.black. “Glutathione metabolism” pathway showed significant enrichment in MM.brown. “Ribosome” pathway showed significant enrichment in MM.green. “Circadian rhythm” and “Glucosinolate biosynthesis” pathways displayed marked enrichment in MM.turquoise. “Selenocompound metabolism” indicated high enrichment in MM.magenta, while “Monoterpenoid biosynthesis” showed enrichment in MM.red (Appendix A).

Combining the correlations between the PCA, heatmap, pathways, and WGCNA, this study revealed that the genes of pink, blue, black, and green modules might play crucial roles in the morphology and quality of kale under UV-A supplementation in different photoperiods.

### 3.9. Validation of DEGs Expression Patterns

To verify the reliability and repeatability of RNA-seq results, quantitative real-time PCR (qRT-PCR) was set to reveal the expression of 15 DEGs that were randomly selected from the top 20 enriched KEGG pathways. The correlation coefficients between RNA-seq results and qRT-PCR were normalized to the mean of the *ACT* gene. Those 15 genes exhibited an observably positive correlation, with 0.9100 < R^2^ < 0.9965, proving that the RNA-seq results were highly reliable, due to the line with the trends in expression detected by qRT-PCR (Appendix A).

## 4. Discussion

Light is an indispensable element for vegetable growth and phytochemical accumulation. Plants perceive light through a variety of photoreceptors to regulate plant growth and development, including blue/UVA photoreceptors: cryptochrome (CRY) and phototropin (PHOT), UVB photoreceptor *UVR8*, and infrared and far-infrared photoreceptor photosensitive pigments (PHY). *CRY1* plays a major role in inhibiting hypocotyl elongation under blue light treatment, whereas *CRY2* mainly inhibits hypocotyl growth under low-intensity blue light (1 μmol·m^−2^ s^−1^) [38]. Under blue light irradiation, *CRY1* interacts with *BZR1*, *BES1*, and *PIF4* to repress target gene expression, thereby inhibiting hypocotyl elongation in Arabidopsis [39]. In this study, the UV-A supplement markedly increased the plant height of kale, with massive increments of stem diameter, plant moisture content, and total leaf area found in DS and LDS (Appendix A). In addition, UV-A supplementations highly increased the contents of chlorophyll a, chlorophyll b, and carotenoids (Appendix A), which resembled the increasing trend of biomass. The PCA results intuitively indicated a positive regulatory relationship between pigment and biomass since the angle between pigment, fresh weight, and dry weight was less than 90° (Figure 3). The kale exposed to supplemental UV-A treatments (6, 12 and 18 μmol·m^−2^ s^−1^) indicated higher contents of chlorophyll a and chlorophyll b than CK, with increases of 18.9%, 13.3%, 17.8%, and 34.5%, 20.7%, and 37.9%, respectively. The highest biomass was found in the 12 μmol·m^−2^ s^−1^ treatment [40]. The 30 min·d^−1^ UV-A (0.5 W·m^−2^) treatment highly promoted the plant height, root length, and total leaf area of *Phaseolus mungo* [41]. Similar results were found in lettuce, with 15~29% dry weight increased by supplemental 10~30 μmol m^−2^ s^−1^ of UV-A. The shoot dry weight (18~32%) and leaf area (15~26%) increased under the treatments of UVA-5d, UVA-10d, and UVA-15d [6]. Daytime supplemental UV-A (31.2 μmol m^−2^ s^−1^) slightly increased the root and total biomass of pea seedlings, while nighttime supplemental UV-A significantly decreased the root and total biomass [11].

Plant hormones are widely recognized as crucial regulatory signals in plants [42]. Auxin (IAA) has an important role in the regulation of plant morphogenesis and growth, which is regulated by a myriad of genes [43]. In tomatoes, *IAA9* will always be the negative regulator of auxin responses. *AS-IAA9* (antisense plants) plants were usually taller than wild-type plants and exhibited enhanced hypocotyl elongation and longer internodes [44]. Compared to CK, 15 μmol m^−2^ s^−1^ blue light treatment significantly increased *GH3.5* expression and, thus, promoted IAA-Asp synthesis activity, which, in turn, controlled pea growth and development [45]. As Figure 4 shown, *IAA27*, *IAA9*, *IAA4*, and four *GH3.12* genes were significantly up-regulated in DS treatment, and both the plant height and stem diameter were significantly increased. Simultaneous supplementation with blue light and UVA (LS and LDS) would co-regulate photoreceptors and, thus, trigger the relevant genes in the plant to inhibit growth hormone signaling, in order to stabilize the growth hormone environment, but when blue light and UVA were treated separately (DS), the expressions of the GH family and IAA family genes were significantly up-regulated, which, in turn, promoted the accumulation of auxin and contributed to morphological changes in kale.

Gibberellin (GA) could promote cell division and expansion and promote the growth of the stem and leaf of plants [46]. The overexpression of *TaCRY1a* reduced plant height and radicle growth in wheat, and *TaCRY1a* interacted with *TaGID1* and *Rht1* to attenuate the *TaGID1-Rht1* interaction. Thus, blue light stimulates *CRY1* by inhibiting *GID1-DELLA* interactions, thereby stabilizing the DELLA proteins and enhancing their inhibition of plant growth [47]. Both *GID1B* and *RGL3* were significantly up-regulated under DS treatment in this study (Figure 4), suggesting that dark-UVA supplementation may have influenced the synthesis of gibberellin and, thus, the morphological changes in kale.

In this study, different light/dark photoperiod supplementation of UV-A (LS, DS, and LDS) stimulated the growth of kale. Plant height, stem diameter, and total leaf area were revealed higher under DS treatment (Figure 2), and their angles were less than 90°, which indicated those indexes were highly correlative (Figure 3). From WGCNA, plant hormone signal transduction pathways were found to be predominantly enriched in MM.pink and MM.blue, while the brassinosteroid biosynthesis pathway was exhibited to apparently enrich in MM.green (Appendix A), Furthermore, the plant height, stem diameter, and total leaf area revealed a strong positive correlation with MM.pink and MM.green (Figure 6D). MM.pink and MM.green were found to have a noteworthily up-expression under DS treatment (Figure 6C). Combined, the results above (PCA, heatmap, and WGCNA), this study demonstrated that the combined effect of blue light and UV-A co-stimulated photoreceptors resulted in increased expression of phytohormone-related genes (*IAA*, *GH3*, *GID1*, *ARR*, etc.), transmitted specific signals and ultimately affected the growth and biomass of kale. However, the simultaneous supplementation of blue light and UVA at the same time (LS and LDS) might cause abiotic stress and, thus, lead to a decrease in morphology, while the UV-A supplementation treatment at the dark photoperiod (DS), where blue light and UV-A stimulated photoreceptors at different period, achieved the maximum superimposed effect, resulting in the highest plant height, stem diameter, leaf area, and specific leaf weight under this treatment. However, plant hormones are synergistically or antagonistically related to each other, and their regulation of growth and morphology is intricate and complex. How UVA regulates complex plant hormone signaling at different supplemental photoperiods and affects the growth of kale needs to be explored more deeply.

Plants have endogenous circadian rhythm functions at the cellular level and a circadian clock that generates 24-h oscillations in gene expression to anticipate diurnal variation of external environment factors and regulates physiological processes in organisms to synchronize with the external light-dark cycle (LDC) [48,49]. *CCA1* and *LHY* genes are morning-expressed *MYB* transcription factors that have overlapping functions. *TOC1* and *PRR1* genes are evening-expressed genes [50]. CONSTANS (CO) is a key transcription factor in light perception and photoperiodic regulation through the circadian clock too [51]. Under light/dark (14 h/10 h) treatment, *ZmCOL06* and *ZmCOL19* were indicated to be down-regulated in the plumule stage of maize, while the radicle and immature leaves stage were up-regulated [52]. In this study, *LHY*, *APRR5*, and two *APRR1* genes exhibited distinct up-regulation in DS, while two *APRR9* genes were inhibited (Appendix A). Meanwhile, seven CONSTANS-like genes (two *COL15*, *COL9*, two *COL10*, *COL6*, and *CIA2*) all revealed prominent up-regulation in DS. Circadian rhythm pathway displayed remarkable enrichment in MM.turquoise (Appendix A), while MM.turquoise presented significant correlations with plant height, stem diameter, total leaf area, total fresh weight, and total dry weight (Figure 6D). Meanwhile, MM.turquoise displayed significant up-expression in DS (Figure 6C), which matched the results with the plant hormone signal pathway. In short, UV-A supplementation in different light/dark photoperiods stimulated the related key genes, boosted LDC, and eventually affected the growth and biomass of kale, while DS treatment generated the most.

Glucosinolates (GLSs) are sulfur-rich and nitrogen-rich secondary metabolites unique to cruciferous species and can be classified as aliphatic glucosinolate (AGS), indolic glucosinolate (IGS), and aromatic glucosinolate (RGS), depending on their amino acid precursors [53]. In this study, seven GSLs, including three aliphatic GLSs and four indolic GSLs, were exacted and detected by HPLC (Figure 1). Generally, UV-A supplementation in different light/dark photoperiods signally increased indolic GLSs and total GLSs content, while LS and LDS treatments significantly increased aliphatic GLSs. Among them, LS remarkedly increased the SIN content, DS treatment significantly increased the GNA and 4ME content, and LDS treatment signally increased the GBS content. The content of 2-phenylethylthioside and 4-methoxy-indole-3-methylenethioside in white mustard under UVA (365 nm) treatment exhibited significantly higher than control [54]. Supplemental 40 µmol·m^−2^ s^−1^ UVA-radiation exposure (under 250 µmol·m^−2^ s^−1^ white light) remarkedly up-regulated *UVR8* and other genes related to the glucosinolate biosynthesis pathway, ultimately promoting the GLS accumulation of Chinese kale [31], highly resembling this study (Figure 4). Glucosinolate biosynthesis pathway consists of three parts: side chain elongation of methionine (Met) and tryptophan (Phe), core structure formation, and side chain modification [55]. In the core structural part, dark-UVA and light/dark-UVA supplementation significantly upregulated the expression of *CYP79B1*, *CYP79B3*, two *CYP79A2*, and *SOT17*. *MYB34*, *MYB51*, and *MYB122* were important transcription factors for indolic GLSs biosynthesis [56]. *ATR1/MYB34* has been verified to activate *CYP79B2*, *CYP79B3*, and *CYP83B1* and regulate homeostasis between indolic GLS and IAA biosynthesis [57]. In this study, *MYB34* (ncbi_106322588) illustrated down-regulation under LS and LDS treatments, while indicating signally up-regulated under DS. The other *MYB34* (ncbi_106303800) exhibited more expression (5.63 < FPKM < 10.273) under DS and LDS treatments than the former *MYB34* (FPKM < 0.983) (Appendix A). Glucosinolate biosynthesis illustrated enrichment in MM.black, MM.magenta, and MM.turquoise (Appendix A), but GNA, 4OH, GBS, 4ME, NEO, and total GLS demonstrated strong correlations with MM.turquoise and MM.black, which were predominantly up-regulation in DS and LDS treatment (Figure 6) and matched the GLS content (Figure 1). Overall, dark-UVA supplementation and light/dark-UVA supplementation accelerated the GLS biosynthesis, especially indolic GLS biosynthesis, by triggering *MYB34* to stimulate the key genes (*CYP79B1*, *CYB79B3*, and *CYP79A2*).

UV-A supplementation in different photoperiods (LS, DS, and LDS) signally promoted the nutritional quality of kale in this study. UV-A supplementation remarkedly increased soluble protein and Vc contents, with a massive reduction of nitrate content, while soluble sugar content barely changed (Appendix A). Lettuce leaves grown under low UV-A intensity (10 µmol·m^−2^ s^−1^) also maintained the soluble protein (secondary respiratory substrate) content during storage, whereas the leaves of the control plants or plants grown under high UV-A intensity (40 µmol·m^−2^ s^−1^) underwent a decrease [12]. Supplemental UV-A (10, 20, 30 µmol·m^−2^ s^−1^, 16 h) treatment exhibited higher contents of soluble sugar, soluble protein, and Vc, about 12.74~26.11%, 13.76~23.53%, and 61.04~66.11%, respectively, compared with lettuce grown under mixed blue, red, and far-red light, with a photon flux density of 237 µmol·m^−2^ s^−1^ in the growth room. Meanwhile, lettuce exposed to different durations (5 d, 10 d, and 15 d) of UV-A revealed higher contents of Vc and soluble sugar, about 47.46~63.19% and 21.69~42.17%, respectively [14]. Supplemental UV-A (6 µmol·m^−2^ s^−1^) signally decreased the nitrite content of red-leaf lettuce, but massively increased in green-leaf lettuce [58]. Vc displayed a positive correlation with soluble protein and a negative correlation with nitrate, which observably matched the results in this study (Figure 3).

Plants could synthesize some antioxidant compounds to cope with UV stresses, e.g., Vc, phenolic, flavonoids, etc. In the present study, UV-A supplementation in different photoperiods (LS, DS, and LDS) yielded higher Vc content of kale. DS and LDS signally promoted FRAP, and LS inhibited total phenolic content, while DPPH and total flavonoids content remained unchanged under supplemental UV-A in different photoperiods of kale (Appendix A). Total phenolic content revealed negative correlations to DPPH and total flavonoids, while FRAP exerted positively correlation with Vc (Figure 3). Pea plants could regulate the epidermal UV-A absorption and accumulation of individual flavonoids by sensing the complex radiation signals extending into the visible region of the solar spectrum [59]. The compounds overproduced by 2 h UV-A (3.16 W·m^−2^) treatment were GAH I, 4-O-CQA, GAD, sinapic acid and 1-S-2, 2-diFG, with increments of ~14, 42, 48, 7, and 61%, as compared to 7-day-old control broccoli sprouts [60]. Supplemental UV-A radiation (40 µmol·m^−2^ s^−1^) dramatically enhanced FRAP (~18.3%) of broccoli microgreens [61]. In response to various stress stimuli in nature, plants produce a range of antioxidants to reduce stress, including glutathione (GSH). *GhGSTF1* and *GhGSTF2* could restore pigmentation in the hypocotyl of the Arabidopsis mutant tt19-7 [62]. The glutathione s-transferase (GST) family influences many redox-dependent processes, including hormonal and stress responses, and plays an important role in cellular metabolism and detoxification [63]. The *AtGSTU7* mutant significantly increased the glutathione content of *Arabidopsis thaliana* compared to the control [64]. In this study, *GSTF10*, *GSTU24*, *GGT1*, *GPX2*, *GSTU7*, *GSTU5*, *GSTF2*, *GSTU17*, *GSTU12*, *GSTU25*, *GSTU11*, and two *GSTF3* were significantly up-regulated in DS and LDS treatments, indicating that dark photoperiod and light/dark photoperiod supplemental with UVA significantly promoted glutathione biosynthesis, which enhanced the antioxidant capacity and increased the antioxidant compounds of kale (Appendix A).

## 5. Conclusions

As a phytochemical-rich vegetable, kale has led to increasing awareness. This study validated the morphological and biometric responses of kale to supplemental UV-A in different light/dark photoperiods via RNA-seq. UV-A supplementation treatments are conducive to yielding greater biomass and accumulating higher GLS content. Moreover, DS treatment remarkedly up-regulated the expressions of the key regulatory genes (*IAA9*, *COL6*, *APRR1*, and *MYB34*, etc.) involved in important pathways (plant hormone signal transduction pathway, circadian rhythm pathway, and glucosinolate biosynthesis pathway, etc.). Thus, supplemental 12 µmol·m^−2^ s^−1^ UV-A in dark photoperiod (DS) might be a valuable method to promote the growth and phytochemical profile of kale in a plant factory.

## Figures and Tables

**Figure 1 antioxidants-12-00737-f001:**
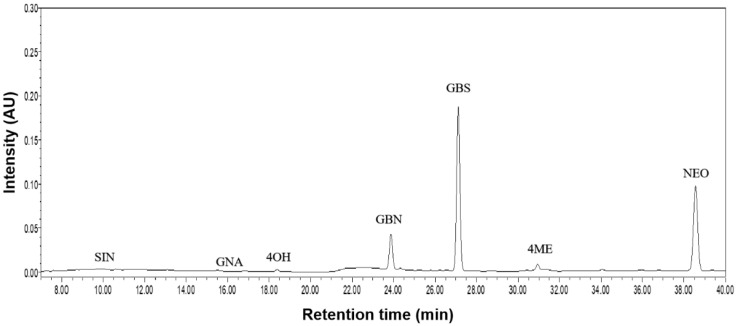
Glucosinolate content of kale under different UV-A treatments. Different lowercase letters reveal significant differences (*p* < 0.05), according to Duncan’s test. Vertical bars indicate the standard margin of error. “LS” = Light-UVA Supplementation, “DS” = Dark-UVA Supplementation, “LDS” = Light/Dark-UVA Supplementation. SIN = sinigrin, GNA = gluconapin, 4OH = 4-hydroxyglucobrassicin, GBS = glucobrassicin, GBN = glucobrassicanapin, NEO = neoglucobrassicin, 4ME = 4-methoxyglucobrassicin, In-GLSs = Indolic Glucosinolate, A-GLSs = Aliphatic Glucosinolates.

**Figure 2 antioxidants-12-00737-f002:**
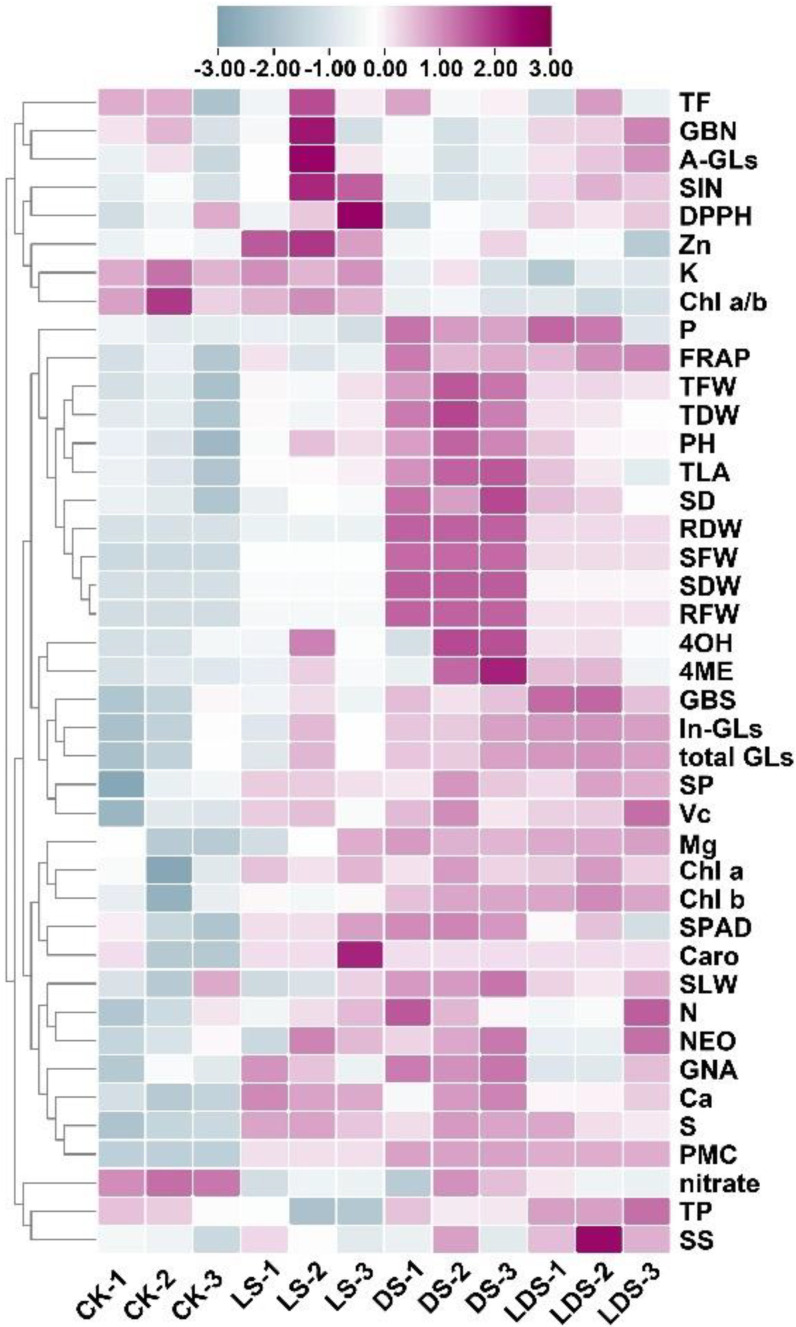
Cluster heatmap analysis summing up the morphology and quality of kale in CK and supplemental UV-A light treatments. A false-color scale with purple as an increased parameter, while blue represents a decreased parameter was used to visualize the results. “LS” = Light-UVA Supplementation, “DS” = Dark-UVA Supplementation, “LDS” = Light/Dark-UVA Supplementation, “PMC” = plant moisture content, “SAW” = specific area weight, “PH” = plant height, “TFW” = total fresh weight, “RDW” = root dry weight, “SD” = stem diameter, “SFW” = shoot fresh weight, “RFW” = root fresh weight, “SDW” = shoot dry weight, “RFW” = root fresh weight, “TDW” = total dry weight, “TLA” = total leaf area, “TP” = total phenolic, “SS” = Soluble sugar, “TF” = total flavonoids, “SP” = soluble protein, “Vc” = Vitamin C, and the full names of all GLS variants has been described in Figure 1.

**Figure 3 antioxidants-12-00737-f003:**
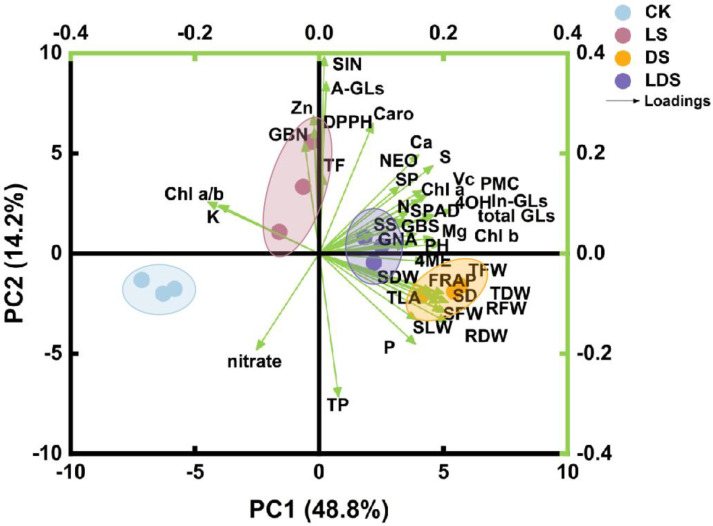
The multivariate principal component analysis presented the relationship between indexes among 4 treatments. The full name has been described in Figure 2. “LS” = Light-UVA Supplementation, “DS” = Dark-UVA Supplementation, “LDS” = Light/Dark-UVA Supplementation.

**Figure 4 antioxidants-12-00737-f004:**
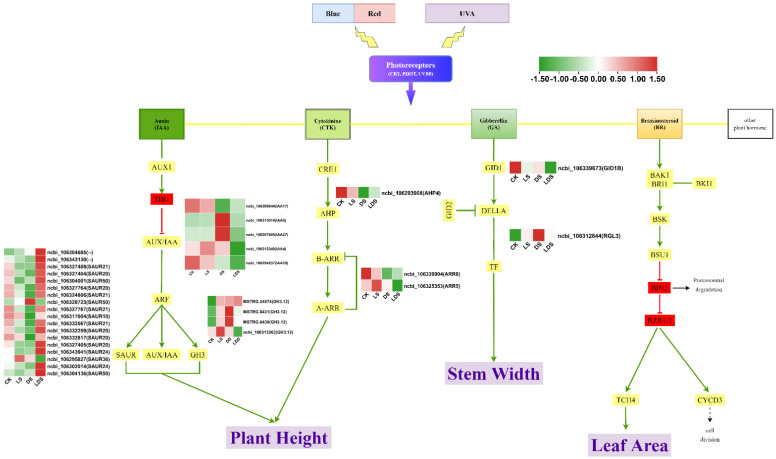
Expression of DEGs on plant hormone signal transduction pathway. Made by diagrams.net.

**Figure 5 antioxidants-12-00737-f005:**
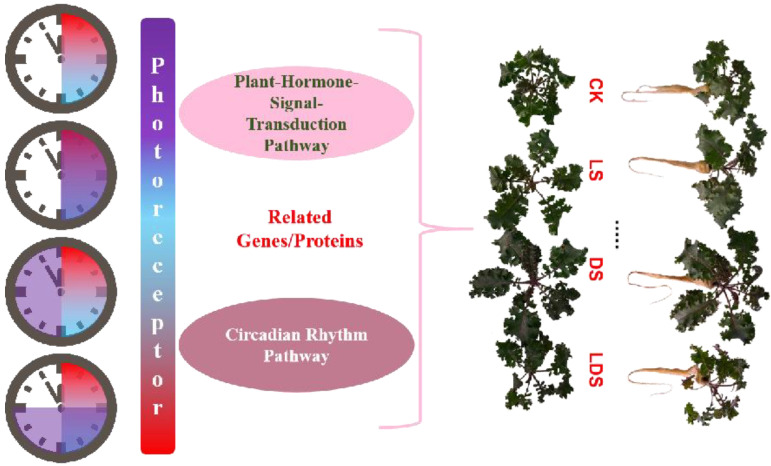
Key regulatory pathways/genes responded to different UV-A supplementation of kale. “LS” = Light-UVA Supplementation, “DS” = Dark-UVA Supplementation, “LDS” = Light/Dark-UVA Supplementation.

**Figure 6 antioxidants-12-00737-f006:**
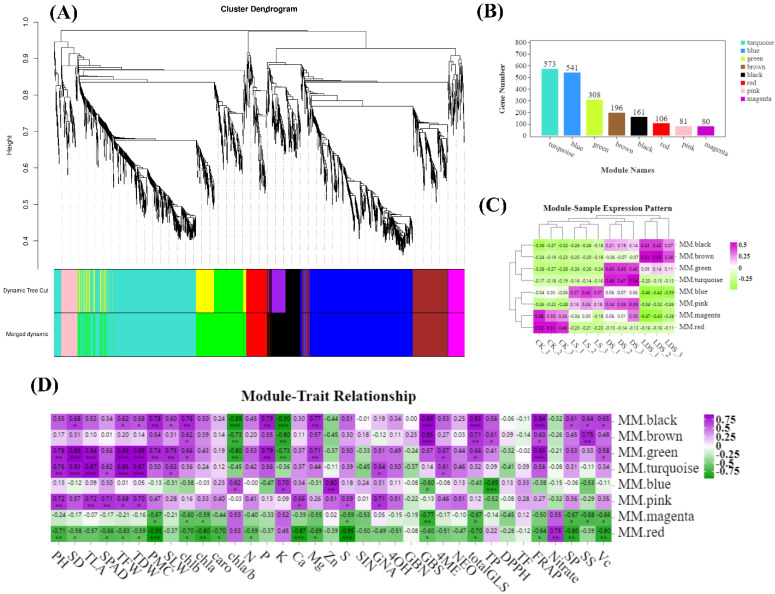
WGCNA of 2047 DEGs. (**A**) Cluster dendrogram indicating 8 modules of co-expressed genes by WGCNA. (**B**) Gene numbers of different modules. (**C**) Module-Sample expression patterns. (**D**) Module-trait correlations. *p* values were shown as: * *p* < 0.05, ** *p* < 0.01, *** *p* < 0.001. The full name of the traits has been described in Figure 2. The color scale shows correlations from positive (purple) to negative (green). “LS” = Light-UVA Supplementation, “DS” = Dark-UVA Supplementation, “LDS” = Light/Dark-UVA Supplementation.

## Data Availability

Data is contained within the article or Appendix A. The raw data presented in this study are available in SRA (accession number: PRJNA940045).

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
