# Peer review of "RNA-Seq Analysis Demystify the Pathways of UV-A Supplementation in Different Photoperiods Integrated with Blue and Red Light on Morphology and Phytochemical Profile of Kale"

_antioxidants, 2023, doi:10.3390/antiox12030737_

Round 1

Reviewer 1 Report

This manuscript has examined biomass amount, phytochemical profile and gene expression of Kale plants under UV-A supplementation and revealed that the supplementation contributes to much amount of biomass and higher GLS content, especially under DS treatment. Authors measured a number of factors and showed that DS treatments would promote growth and phytochemical accumulation. However, this manuscript feels not quite complete. Here, I raised some points to be addressed or corrected before publication as below.

Throughout the text, there are terribly many English grammatical errors which must prevent publication. Some sentences lack a subject. Authors should use any English proofreading services.

Explanations for many abbreviations such as PAR (line 34), FRAP (line 291) and others are not shown. In particular, “LS, DS and LDS” (line 10) suddenly appear in abstract. The abstract must be understood for what it says just by reading it.

Authors should deposit RNA-seq raw and processed data into any public database like SRA and provide the accession number. In addition, provide the information of how to construct RNA-seq libraries and how many replicates of RNA-seq per a condition were performed in “Materials and Methods” section.

GO and KEGG analysis may be done for upregulated and downregulated genes separately, but not for all DEGs.

Lines 360 and 369. “Figure 4” is wrong. “Figure 3” is correct.

“Discussion” section is too terrible. It looks a review article. There, authors should describe the relationship between the obtained data and the previous findings, and the possibilities that can be derived. The description for phytohormones (lines 530-592) should be re-summarized in very shorter form, because authors did never measure and compare amounts of any hormones. Similarly, the description for circadian clock (lines 615-652) should be shortened.

Author Response

 Our deepest gratitude goes to you for your compelling work and thoughtful suggestions that have helped improve this paper substantially. Please find attached the file for our precise responses.

Reviewer 2 Report

The manuscript by Jiang et al. is devoted to an important problem: the study of the mechanisms of influence of UV-A on the growth and phytochemical profile of kale plants. This work contains very interesting results, but I have some remarks about the presentation:

1. P. 1, line 31: “(Jiao et al., 2007; Shi et al., 2018)” should be “[1,2]”.

2. P. 2, lines 79-82: This sentence must be rewritten. Something like: “… to fathom the effects of the supplemental UV-A on the morphology and quality …”.

3. P. 3, line 110: broccoli? Not kale?

4. It seems to be better to add the Figures S1-S5 to the main document, not to the supplementary files.

5. It is necessary to check and correct the legends for all figures in the main document and supplementary. Please check their design and sufficiency of information. Especially Figures 1, 4 and S13 (please, complete the legends).

Author Response

 Our deepest gratitude goes to you for your careful work and predominant suggestions that have helped improve this paper substantially. Please find attached the file for our precise responses.

Round 2

Reviewer 1 Report

Please don't forget to provide your accession number in the published version.

Author Response

Our deepest gratitude for your professional suggestions. All accession numbers of 16 genes used for verifying the expression profiles have been supplemented in Table S1(supplementary material). And the accession number of SRA could be found on line 193. Sincere gratitude for optimizing and accelerating this manuscript’s publication with your professional and compelling suggestions.
